# Field-induced bound-state condensation and spin-nematic phase in SrCu$_2$(BO$_3$)$_2$ revealed by neutron scattering up to 25.9 T

Ellen Fogh [1,12] ✉, Mithilesh Nayak [2,12] ✉, Oleksandr Prokhnenko [3], Maciej Bartkowiak [3,4], Koji Munakata [5], Jian-Rui Soh [1], Alexandra A. Turrini [1,6], Mohamed E. Zayed [7], Ekaterina Pomjakushina [8], Hiroshi Kageyama [9], Hiroyuki Nojiri [10], Kazuhisa Kakurai[5], Bruce Normand[1,11], Frédéric Mila [2] & Henrik M. Rønnow [1]

In quantum magnetic materials, ordered phases induced by an applied magnetic field can be described as the Bose-Einstein condensation (BEC) of magnon excitations. In the strongly frustrated system SrCu$_2$(BO$_3$)$_2$, no clear magnon BEC could be observed, pointing to an alternative mechanism, but the high fields required to probe this physics have remained a barrier to detailed investigation. Here we exploit the first purpose-built high-field neutron scattering facility to measure the spin excitations of SrCu$_2$(BO$_3$)$_2$ up to 25.9 T and use cylinder matrix-product-states (MPS) calculations to reproduce the experimental spectra with high accuracy. Multiple unconventional features point to a condensation of $S = 2$ bound states into a spin-nematic phase, including the gradients of the one-magnon branches and the persistence of a one-magnon spin gap. This gap reflects a direct analogy with superconductivity, suggesting that the spin-nematic phase in SrCu$_2$(BO$_3$)$_2$ is best understood as a condensate of bosonic Cooper pairs.

Condensation of a macroscopic number of particles into a single state is a purely quantum mechanical phenomenon exhibited by a wide spectrum of bosonic systems[1]. Early and spectacular examples of this Bose-Einstein condensation (BEC) include superfluidity in liquid $^4$He and superconductivity in metals, where attractive interactions allow the electrons, which are fermions, to form Cooper pairs, which are bosons. Field-induced gap closure and magnetic order in quantum disordered magnetic materials is conventionally described as a BEC of magnons, which are spin-1 excitations[2,3]; examples of this phenomenon include the dimerized quantum antiferromagnets TlCuCl$_3$[4,5] and BaCuSi$_2$O$_6$[6–8] and, under quite different experimental conditions, yttrium iron garnet[9,10].

By contrast, the field-induced phase diagram of the highly frustrated quantum magnet SrCu$_2$(BO$_3$)$_2$ (Fig. 1a)[11] shows neither clear gap closure nor induced magnetic order, but instead a spectacular series of magnetization plateaux[11–16]. SrCu$_2$(BO$_3$)$_2$ presents a remarkably faithful

$^1$Laboratory for Quantum Magnetism, Institute of Physics, Ecole Polytechnique Fédérale de Lausanne (EPFL), CH-1015 Lausanne, Switzerland. $^2$Institute of Physics, Ecole Polytechnique Fédérale de Lausanne (EPFL), CH-1015 Lausanne, Switzerland. $^3$Helmholtz-Zentrum Berlin für Materialien und Energie, D-14109 Berlin, Germany. $^4$ISIS Neutron and Muon Source, Rutherford Appleton Laboratory, Harwell OX11 0QX, UK. $^5$Neutron Science and Technology Center, Comprehensive Research Organization for Science and Society (CROSS), Tokai, Ibaraki 319-1106, Japan. $^6$Laboratory for Neutron Scattering and Imaging, Paul Scherrer Institute, CH-5232 Villigen-PSI, Switzerland. $^7$Department of Physics, Carnegie Mellon University in Qatar, Education City PO Box 24866 Doha, Qatar. $^8$Laboratory for Multiscale Materials Experiments, Paul Scherrer Institute, CH-5232 Villigen PSI, Switzerland. $^9$Graduate School of Engineering, Kyoto University, Nishikyo-ku, Kyoto 615-8510, Japan. $^{10}$Institute for Materials Research, Tohoku University, Sendai 980-8577, Japan. $^{11}$Laboratory for Theoretical and Computational Physics, Paul Scherrer Institute, CH-5232 Villigen-PSI, Switzerland. $^{12}$These authors contributed equally: Ellen Fogh, Mithilesh Nayak. ✉e-mail: ellen.fogh@epfl.ch; mithilesh.nayak@epfl.ch

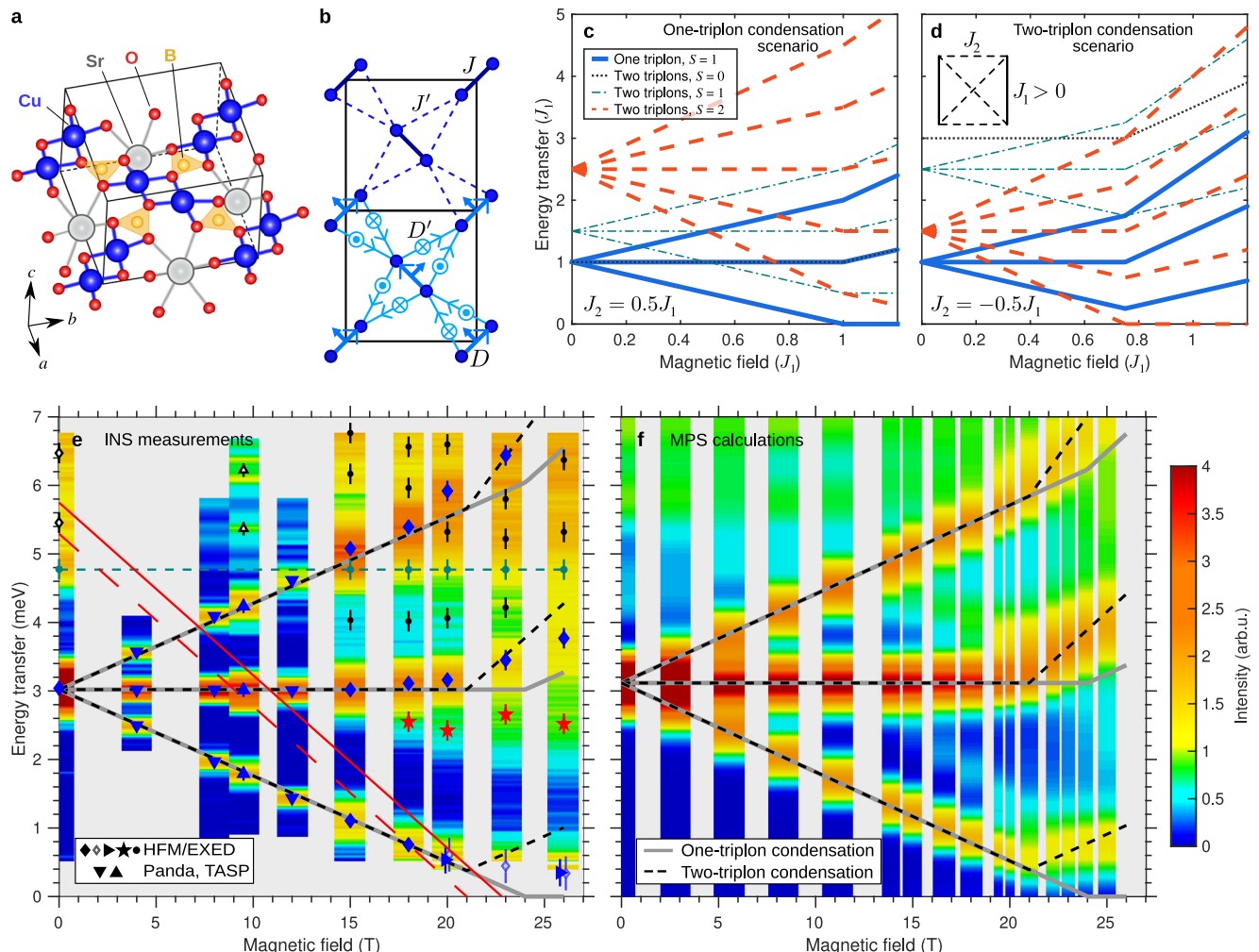

**Fig. 1 | SrCu₂(BO₃)₂ and the magnetic excitation spectrum under field-induced condensation. a** Crystal structure of $SrCu_2(BO_3)_2$ showing one layer of $Cu^{2+}$ ions (half a unit cell along $\hat{c}$). **b** Representation of the superexchange couplings within ($J$) and between ($J'$) the Cu-Cu dimers, which reproduce the Shastry-Sutherland model (SSM), and of the corresponding Dzyaloshinskii-Moriya (DM) interactions ($\vec{D}$ and $\vec{D}'$). **c, d** Energy levels of a $J_1$-$J_2$ two-dimer cluster (inset in panel d) as a function of magnetic field. For $J_2/J_1 > 0$ (**c**) the $t_+$ one-triplon branch condenses first, whereas when $J_2/J_1 < 0$ (**d**) the two-triplon bound state condenses first. The two scenarios result in distinctively different gradients after condensation. **e** Excitation spectra measured by inelastic neutron scattering (INS). Data at intermediate fields (4 – 12 T) were measured at **Q** = (1.5, 0.5, 0), as summarized in the Methods section. Data from our HFM/EXED measurements (0 T and ≥15 T) are presented with full momentum integration, as detailed in Sec. S1 of the Supplementary Information (SI). Colour contours represent the neutron intensity and symbols show the

fitted positions of individual excitations (Fig. 2g–l) with the uncertainties in the mode positions estimated from the corresponding fitting errors. Solid grey lines show the locations of the one-triplon branches in $SrCu_2(BO_3)_2$ for a scenario of one-triplon condensation and no DM interactions (analogous to panel c), dashed black lines for two-triplon condensation (analogous to panel d). The dashed green line shows the $S_z = 0$ branch of the $S = 1$ multiplet within the two-triplon bound state. The solid red line up to 15 T shows the lowest $S_z = 2$ branch of the $S = 2$ multiplet observed by electron spin resonance (ESR)[33], and its extension to higher fields is our linear extrapolation. The dashed red line marks the estimated position of the absolute lowest $S_z = 2$ branch, based on our observation that the change in the one-triplon gradient occurs at 21 T. **f** Corresponding total momentum-integrated spectral functions obtained from MPS calculations on the SSM for $SrCu_2(BO_3)_2$, meaning including DM interactions; overlaid are the solid grey and dashed black lines from panel e (i.e. without DM interactions).

realization of the Shastry-Sutherland model (SSM), which is a paradigm for ideal frustration in a two-dimensional (2D) spin system[17]. This system consists of $S = 1/2$ dimer units arranged orthogonally on a square lattice (Fig. 1b), and its properties are governed by the ratio between the intra-dimer interaction, $J$, and inter-dimer one, $J'$. For $J'/J < 0.675$, the ground state is an exact dimer-product singlet state[18], and the ideally frustrated geometry leads to many unconventional effects, including near-dispersionless one-magnon excitations (to which we refer henceforth as triplons) and very strongly bound multi-triplon states[19–22]. At ambient pressure and zero field, $SrCu_2(BO_3)_2$ is very well described by a SSM with $J'/J \simeq 0.63$[23] and shows anomalous thermo-dynamic behaviour[24–27] due to the high spectral density of highly localized spin excitations. Under an applied hydrostatic pressure, its behaviour mirrors the phases of the SSM as $J'/J$ is changed[27–29].

Most remarkable of all is the appearance of field-induced mag-netization plateaux in $SrCu_2(BO_3)_2$: extremely strong magnetic fields applied along the $c$-axis (Fig. 1a) induce plateaux at 1/8, 2/15, 1/6, 1/4, 1/3, 2/5 and 1/2 of the saturation magnetization, which is reached around 139 T[16]. In direct analogy with charge-density-wave (CDW) order in electronic systems, these plateaux can be seen as bosonic CDW phases of magnetic entities. The predicted spin superstructure on the lowest (1/8) plateau is a crystal of $S = 2$ two-triplon bound states, as opposed to a crystal of unbound $S = 1$ triplons[30]; this prediction awaits experi-mental verification. Although the transition to the 1/8 plateau takes place with a jump at $\mu_0 H_c = 27$ T, the magnetization is clearly finite above 16 T, and above 21 T it increases linearly until the jump[14,31]. In this crossover region, the gap to the lowest triplon branch does not close as the field is increased, but shows an avoided crossing previously

ascribed to the presence of weak Dzyaloshinskii-Moriya (DM) interactions[32–34].

It was suggested very early that the field-induced gap closure in $SrCu_2(BO_3)_2$ could be anomalous[35]. The zero-field gap of the nearly immobile triplons is $\Delta \simeq 3.0$ meV, but triplon pairs gain kinetic energy from correlated hopping, such that two-triplon states with $S = 0$ and 1 lie far below $2\Delta$ and hence are strongly bound (with respective energies $4.0$[36] and $4.8$ meV[19]). For the $S = 2$ multiplet, third-order perturbative calculations on the SSM suggest that its lowest branch, which is doubly degenerate and reaches its minimum at momentum $(\pi, \pi)$, also lies below $2\Delta$[37]. This state is further characterized by the twofold-degenerate irreducible representation $E$ of the point group to which it belongs. However, calculations by a variational method suggest that qualitative changes set in beyond $J'/J \approx 0.4$ and a different spin-2 bound state, which has $A_2$ symmetry, a nondegenerate irreducible representation of the point group and is minimal at momentum $(0, 0)$, becomes lower[38]. In either case, increasing the field should cause a $S = 2$ bound state to meet the singlet ground state before the lowest one-triplon branch does, resulting not in field-induced magnetic order, but instead a BEC of two-triplon bound states, which is a spin-nematic phase[39,40]. However, a direct demonstration of spin-nematicity in $SrCu_2(BO_3)_2$ has not to date been possible: standard momentum-resolved experimental probes do not observe $S = 2$ excitations, while light-scattering methods probe them only at momentum $\mathbf{Q} = (0, 0)$ and for specific point-group symmetries, excluding both the $E$ and $A_2$ states.

The definitive experiment would instead be indirect, determining the dynamical structure factor at all $\mathbf{Q}$ by inelastic neutron scattering (INS) and comparing this with model calculations. For a conceptual illustration of this statement, we use Fig. 1c, d to contrast the two scenarios in a model consisting of a two-dimer cluster with fully frustrated interdimer coupling that we discuss further below. These figures show how the behaviour of the one-triplon excitations is quite different in the two scenarios of $S = 1$ condensation ($J_2 > 0$ in the two-dimer model) and $S = 2$ condensation ($J_2 < 0$), providing an indirect but unambiguous probe of the relevant physics. Although INS has historically been limited to fields below 15 T, for a limited period neutron scattering was possible up to 25.9 T (static field) on a specialized facility at the Helmholtz-Zentrum Berlin (HZB)[41–44], thus enabling the investigation of $SrCu_2(BO_3)_2$ over most of the formerly dark field regime below $H_c$. Model calculations for frustrated 2D quantum systems have also been a historical impossibility, but recent progress in numerical methods based on matrix-product states (MPS) has made it possible to calculate hitherto unavailable spectral functions with quantitative accuracy. These parallel developments render high-field INS the ideal tool to decode the spin dynamics in $SrCu_2(BO_3)_2$ and in a range of other systems[8].

Here we harness this progress to demonstrate compelling evidence for bound-state condensation forming a spin nematic in $SrCu_2(BO_3)_2$. We measure the excitation spectrum using INS at 200 mK with magnetic field strengths up to 25.9 T (Fig. 1e), and we perform MPS calculations of the dynamical structure factor of the SSM with weak DM interactions (Fig. 1f). By tracking the Zeeman splitting of the one-triplon mode, we find field-induced changes in the gradients of the triplon branches and a persisting, DM-independent gap that are fully consistent with bound-state condensation. Our INS and MPS spectra also reveal a rapid, field-induced transfer of weight to multiple novel excitations, which we show are two- and three-triplon composites that can only exist if the condensate contains bound states.

## Results
### Inelastic neutron scattering
We performed INS experiments at the High-Field Magnet (HFM) facility that operated recently at the HZB. The time-of-flight EXED instrument offered the unique capability of performing neutron diffraction and

spectroscopy at fields up to 25.9 T. One of the compromises inherent to working at these fields is a rather limited reciprocal-space coverage, as explained in Sec. S1A of the Supplementary Information (SI), where we provide a detailed description of the HFM/EXED experimental geometry. Unless otherwise stated, all the INS data we show were collected with incoming energy $E_i = 8$ meV, yielding an energy resolution of 0.64(1) meV. A further key to our experiments was a dilution refrigerator enabling a sample temperature of 200 mK, well below the smallest energy scales in $SrCu_2(BO_3)_2$. Further information concerning our measurements is summarized in the Methods section, and in Sec. S1B of the SI we describe our data preprocessing and background subtraction.

Figure 1e shows fully $\mathbf{Q}$-integrated neutron intensities measured as a function of applied field and energy transfer. In Fig. 2a–f we show the neutron intensity measured as a function of $Q_h$, with partial integration around $Q_k = 0$. One of the special properties of the ideally frustrated SSM geometry is that the excitation energies are almost entirely independent of $\mathbf{Q}$, and indeed all of the excitations we observe in $SrCu_2(BO_3)_2$ are rather flat over the full extent of the accessible reciprocal space; only their intensity increases monotonically with $Q_h$. This lack of dispersion justifies integrating the intensity data over a large $\mathbf{Q}$ volume, which improves the counting statistics significantly, albeit at the cost of picking up an additional contribution to the line-width from the minor dispersion that is present (Fig. 2a–f). The intensity in every branch is concentrated at the largest values of $Q_h$ for all fields, while the relative intensities between branches show weak changes with $Q_h$ (which we analyse in Sec. 1D of the SI).

Integrating over the entire available $\mathbf{Q}$ range leads to the intensities shown in Fig. 2g–l; the fitting procedure to multiple Gaussians is described in Sec. 1C of the SI. At zero field (Fig. 2g), the one-triplon excitation is found around 3.0 meV, with a number of higher-lying excitations at 4.8, 5.5 and 6.5 meV, all in good agreement with previous INS measurements[19–22]. In an applied field, the one-triplon excitation shows Zeeman splitting into three branches, which are clearly visible at 15 T (Fig. 2h). We label the bottom, middle and top branches respectively as $t_+$, $t_0$ and $t_-$. Their splitting increases linearly with the field until 18 T (Fig. 2i), as also observed by ESR[33]. Above this field, the $t_+$ branch curves towards a rather constant value around 0.5 meV, suggesting an avoided crossing, with no indication that the one-triplon gap closes. In addition, we observe the development of a shoulder on the low-energy side of the $t_0$ triplon, which becomes progressively stronger as the field is raised (Fig. 2i–l). Around the $t_-$ triplon we observe further intensity contributions over a broad energy range, which also become stronger at high fields and can be fitted reasonably by three separate Gaussians. Within our experimental resolution, the energies of these new excitations remain largely constant as the magnetic field is increased (Figs. 1e and 2i–l).

As the two-dimer cluster of Fig. 1c, d showed, the field-induced evolution of the lowest ($t_+$) triplon mode is important for identifying the nature of the condensation process. In our experiments, the fact that its gap remains finite throughout the crossover regime improves the prospects for resolving its position close to the elastic line. We performed measurements with an incoming energy of $E_i = 4$ meV, at which the resolution improves to 0.24(1) meV. In Fig. 3a we show the neutron spectra measured at 20 and 25.9 T, where a coherent low-energy feature remains discernible, and we extract the respective positions 0.54 and 0.35 meV (also marked in Fig. 1e), reinforcing our result that the one-triplon gap does not close at any field.

To analyse our intensity data in more detail, in Fig. 3b we show the spectral weights accumulated at energy transfers above a 0.7 meV cut-off for each measurement field. At zero field, this weight is zero until the 3.0 meV triplet branch is encountered, after which it is constant again until the $S = 1$ branch of the two-triplon bound state at 4.8 meV, and beyond this it increases linearly as multiple higher-lying states are encountered. At 15 T, the one-triplon sector is split into three clearly

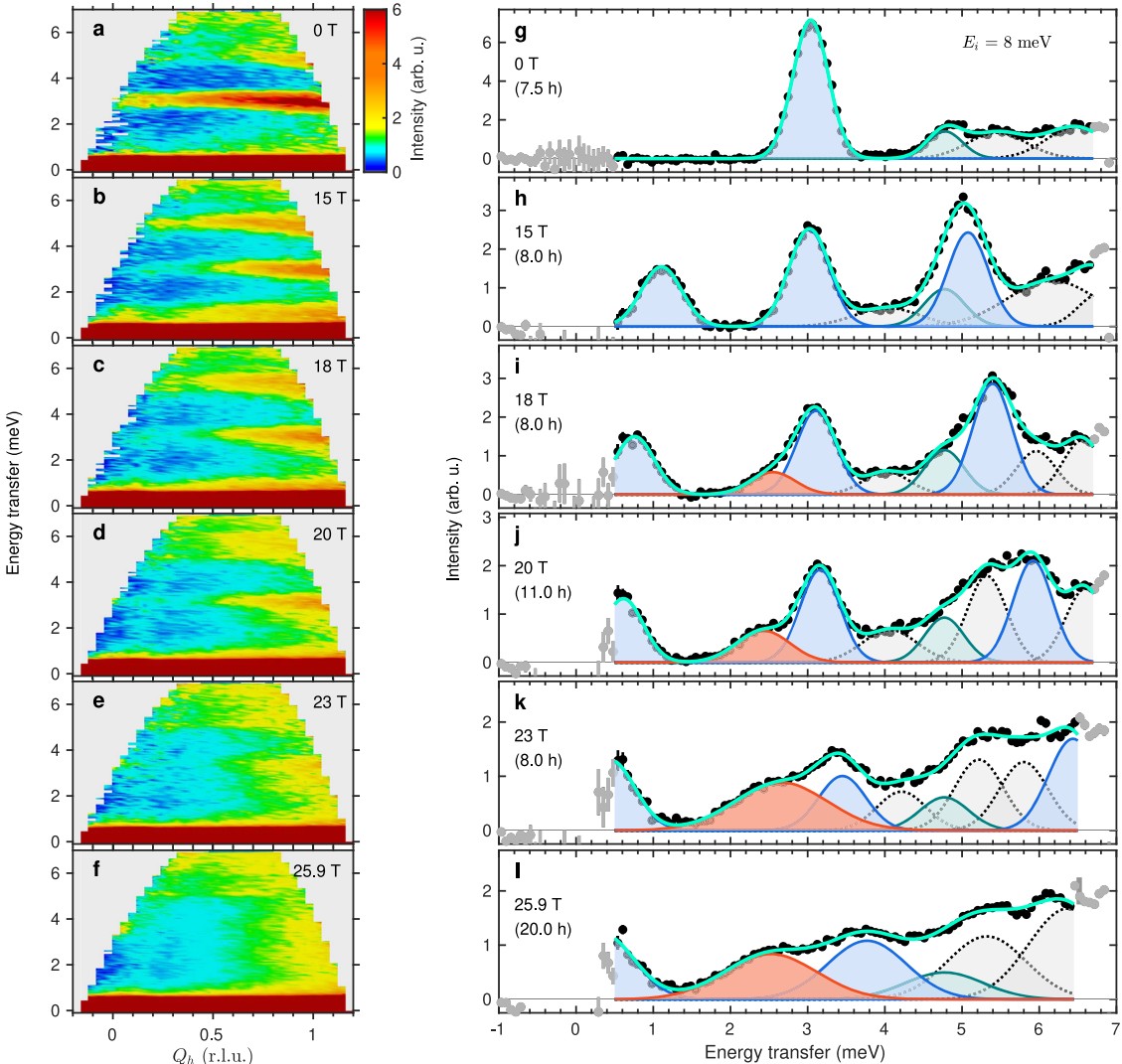

**Fig. 2 | Field-induced evolution of the excitation spectrum in SrCu₂(BO₃)₂.**
**a–f** Measured neutron intensity (colour contours) shown as a function of energy transfer and of momentum $Q_h$ for six different field strengths. These data were integrated over the intervals $Q_k = [-0.75, 0.75]$ and $Q_l = [-1.5, 2.5]$ and no background was subtracted. It is clear that all excitations have very little dispersion, justifying the integration of our data over a large region of reciprocal space.
**g–l** Neutron intensity obtained by further integration over $Q_h = [-0.25, 1.25]$ (black symbols), shown as a function of energy transfer at the same six field strengths. A single background was subtracted as detailed in Sec. S1B of the SI. The one-triplon excitation (solid blue lines with blue shading) exhibits Zeeman-type splitting in the

applied field and its branches were fitted by related Gaussians (Sec. S1C of the SI). The $S_z = 0$ branch of the $S = 1$ two-triplon bound state (dark turquoise lines and shading) appears at a constant energy. Additional intensity develops below the $t_0$ one-triplon branch at $\mu_0 H \geq 18$ T (solid red lines with red shading) and was fitted with a single Gaussian. Further additional intensity visible around and above the $t_-$ one-triplon branch (dotted black lines with grey shading) was fitted with multiple Gaussians. Uncertainties on the neutron counts, $N$, is $\sqrt{N}$ (Poisson counting statistics) and errors in panels (**g–l**) are propagated from the neutron counts taking into account all normalization factors.

defined branches, each of which gives a significant jump in the accumulated spectral weight. At higher fields, these jumps become increasingly broad as additional states appear at multiple different energies, particularly those directly below the $t_0$ and $t_-$ branches (Figs. 1e and 2i–l). We defer an interpretation of these results (Fig. 3c) to our numerical analysis below. To investigate the physics of the different spectral-weight contributions identified in Fig. 2g–l, in Fig. 3d we inspect their evolution as a function of the field. It is clear that the intensity of the additional states (i.e. those below $t_0$ and below $t_-$) scales linearly beyond the crossover region, meaning that it follows the magnetization, $m$, while the one-triplon weight is proportional to $1 - m$.

### Numerical modelling
To interpret the experimental results, we have performed cylinder MPS calculations of the dynamical spectral function of SrCu₂(BO₃)₂. Following multiple previous studies, we model the system using the

SSM with $J'/J = 0.63$ supplemented by the weak DM interactions $D$ and $D'$, which were deduced from experiment[34] to be respectively 3% of $J$ and of $J'$. To illustrate the broadest features of the spectrum we appeal to the two-dimer model of Fig. 1c, d and for a detailed identification of the excitation types we perform exact diagonalization (ED) calculations for the SSM on clusters of $4 \times 4$ spins. A summary of our MPS calculations is provided in the Methods section, and in Sec. S2A of the SI we present additional details including the effects of the MPS cylinder width, which for most of the results we show was limited to $W = 4$. MPS results corresponding to the INS spectrum are shown in Fig. 1f, which makes clear that the calculations contain all the features found in the measurements, most notably the additional excitations appearing below the $t_0$ and $t_-$ one-triplon branches beyond 18 T, at energy transfers in semi-quantitative agreement with experiment.

The first striking feature of our MPS results is the minimum in the $t_+$ branch at 21 T, with a finite gap of 0.5 meV, beyond which the mode

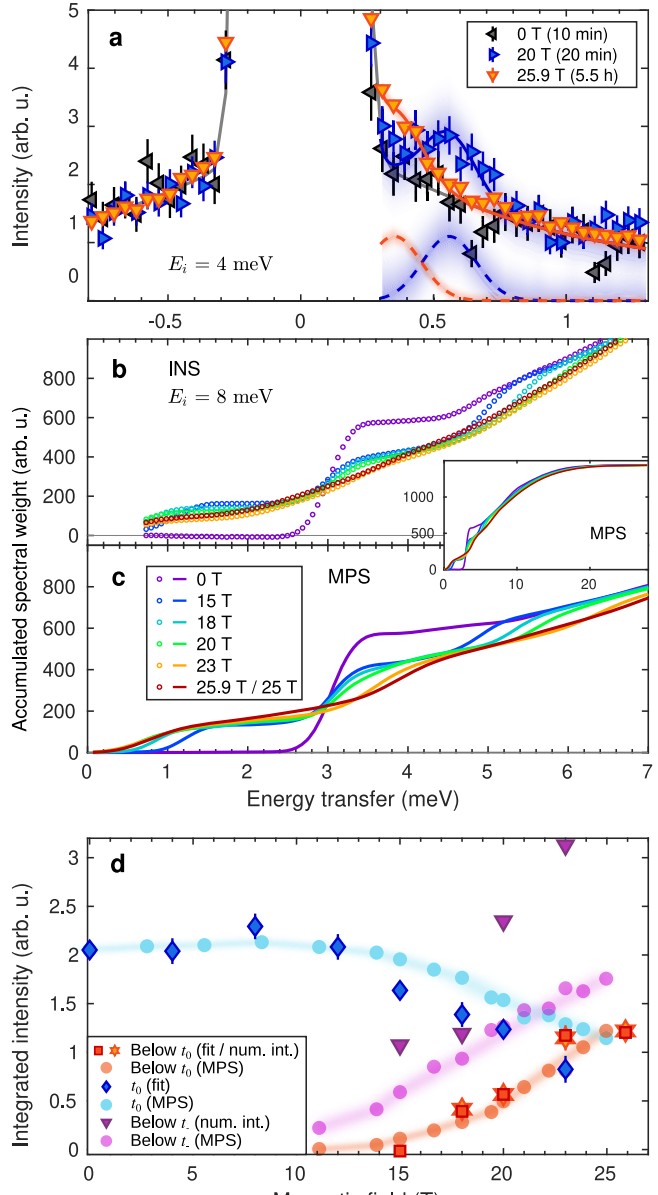

**Fig. 3 | Low-energy INS data and comparison of INS and MPS results. a** Data collected at 0 (black), 20 (blue) and 25.9 T (red) with incident energy $E_i = 4$ meV to resolve the low-energy excitations (solid blue and red lines). Integration range and background determination (solid black line) are detailed in Sec. S1B of the SI. Uncertainties on the neutron counts, $N$, is $\sqrt{N}$ (Poisson counting statistics) and errors here are propagated from the neutron counts taking into account all normalization factors. Mode positions were determined by fitting to a single Gaussian (dashed blue and red lines). The shading indicates the uncertainty on the fits. **b, c** Accumulated spectral weight as a function of energy transfer at each field for our INS (**b**) and MPS data (**c**). The inset shows an extended energy range calculated by MPS. **d** Integrated intensity of different low-energy spectral contributions as a function of field for both the INS and MPS data. Error bars on integrated intensities are derived from fitting errors on peak widths and intensities.

energy increases again with the field. These low energies were difficult to access on HFM/EXED because of the elastic peak (Fig. 3a), but our MPS calculations confirm the absence of low-energy spectral weight here. The persistence of a gap is not a generic feature of field-induced magnon BEC[2,3], where the gap closes, the lowest magnon condenses and spectral weight appears down to zero energy at all higher fields. This spectral weight should extend up to the magnon bandwidth,

which in SrCu₂(BO₃)₂ is only 0.3 meV[45], and thus a minimum around 0.5 meV suggests different physics.

It was assumed previously that this avoided closure is a consequence of the DM interactions in $SrCu_2(BO_3)_2$, and indeed DM terms are known to preserve a gap in systems undergoing magnon BEC[46]. However, the minimum should then appear at the field where one-magnon condensation occurs, which by extrapolating both our INS and MPS results (Fig. 1e–f) is clearly 24 T rather than the value $\mu_0 H' = 21$ T that we observe. To confirm that this gap is not caused by the DM interactions, we repeated our MPS calculations with different values of $D$ and $D'$, as we show in Sec. S2B of the SI. A robust gap exceeding 0.45 meV remains if the DM interactions are removed, and hence we establish that it is an intrinsic property of the pure (Heisenberg) SSM.

The natural explanation is the condensation of the $S = 2$ bound state. As noted above, the two-dimer cluster (inset, Fig. 1d) is sufficient to illustrate some very clear effects in the one-triplon branches. With antiferromagnetic intradimer interactions ($J_1 > 0$), the field-induced condensation process is readily controlled using the interdimer bonds: if $J_2 > 0$, the $t_+$ branch condenses (Fig. 1c) as in a conventional magnon BEC, whereas for $J_2 < 0$, the lowest branch of the two-triplon multiplet condenses (Fig. 1d). In the latter case, the one-triplon branches show several distinctive features: (i) the minimum energy of the $t_+$ branch is half of the binding energy, and the 0.5 meV observed by MPS is consistent with half the binding energy predicted by perturbation theory and iPEPS calculations[30]; (ii) this minimum occurs at a field, $H'$, lower than the extrapolated $t_+$ condensation field; (iii) beyond $H'$, the $t_+$ energy increases linearly with the field; (iv) the $t_0$ branch has a kink at $H'$ and increases with a gradient that is twice as large; (v) the $t_-$ branch also has a kink at $H'$ and increases with a gradient three times as large. These observations are completely generic for the condensation of $S = 2$ bound states and are fully consistent with both the experimental and the numerical results (Fig. 1e, f).

Turning now to the novel field-induced excitations, we consider first their spectral weight. In Fig. 3c we show the accumulated spectral weight computed by MPS in the same format as the experimental results of Fig. 3b, also extending the MPS results to higher energies (inset). The agreement is excellent, with both datasets reflecting the field-induced smoothing of the steps as increasing amounts of weight are shifted to new excitations that appear at intermediate energies. By benchmarking the total weight at high energies, it is clear that the spectral weight lost from the INS energy window (0.7–7.0 meV) is pushed lower (Fig. 3a). To quantify these weight-shifts, in Fig. 3d we show the MPS intensities integrated over energy ranges spanning the $t_0$ branch, below this branch and below the $t_-$ branch. Beyond the obvious agreement with experiment, the MPS results confirm the linear growth of weight in multi-triplon excitations with fields beyond 16 T and the remarkable (2/3) loss of one-triplon weight over the field range from there to the 26 T edge of the critical region around $H_c$.

To go further in identifying these multi-triplon states, it is useful to separate the spectral function into its longitudinal [$S^{zz}(\omega, H)$] and transverse [$S^{xx}(\omega, H)$] components, which we show in Fig. 4a, b. The $t_0$ branch appears in the $S^{zz}(\omega, H)$ channel and the $t_\pm$ modes in the $S^{xx}(\omega, H)$ channel, as we observe at low fields. However, above 21 T there is a rapid growth of low-energy spectral weight in $S^{zz}(\omega, H)$, specifically over a broad range centred near 0.7 meV, and of weight around the $t_0$ energy in $S^{xx}(\omega, H)$. In Fig. 4c, d we show the energies and intensities of selected high-weight states obtained around $H'$ in our cluster ED calculations for the SSM (meaning with no DM interactions), which as detailed in Section S2D of the SI allow us to make an unambiguous identification of the nature ($S$ and $S^z$ quantum numbers) of these states. It is clear that two-triplon composite states appear at low energies in both sectors and that many three-triplon composites are present at energies extending down to the $t_0$ branch.

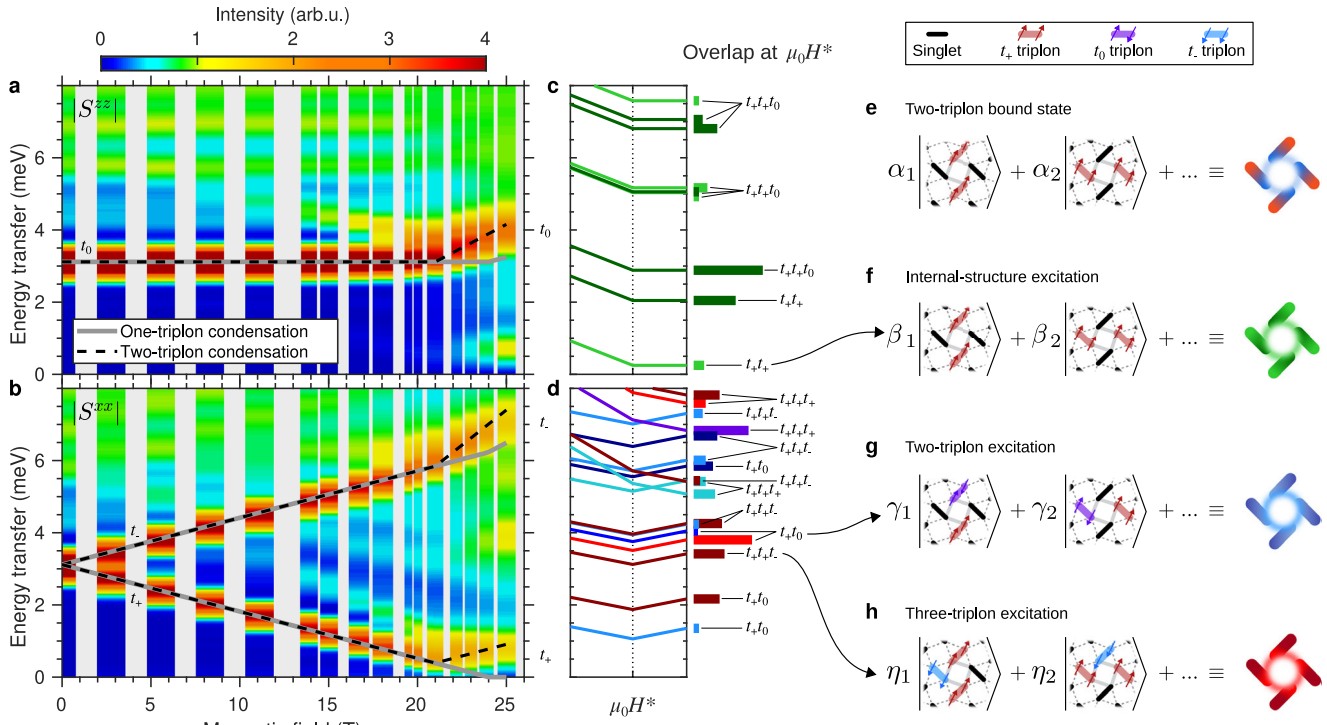

**Fig. 4 | Interpretation of the MPS spectrum. a, b** Spectral functions $S^{zz}(\omega, H)$ and $S^{xx}(\omega, H)$ obtained by cylinder MPS calculations and compared with the one-triplon branches obtained from the one- and two-triplon-condensation scenarios represented in Fig. 1. **c, d** Energies and ground-state overlaps of states obtained in illustrative ED calculations performed around $H^*$ on a $4 \times 4$ cluster. **e–h** Schematic representations of the pinwheel structure of the two-triplon bound state (**e**), an internal excitation of the pinwheel (**f**) which retains the $t_+t_+$ character, a two-triplon excitation consisting of a $t_+$ and a $t_0$ (**g**) and a $t_+t_+t_-$ three-triplon excitation (**h**). In each case illustrated, the coefficients of the basis states shown have the same amplitude and differ only by their phases.

The low-energy spectral weight in $S^{zz}(\omega, H)$ (Fig. 4a and c) is another strong statement for BEC of bound states: the operator $S^z$ can excite either a singlet to a $t_0$ triplet, at relatively high energy, or a transition inside the band of condensed excitations. As noted above, one-triplon condensation in SrCu$_2$(BO$_3$)$_2$ would be accompanied by weight at and below the triplon bandwidth, which at 0.3 meV is too small to explain the MPS result. By contrast, the internal structure of the $S = 2$ bound state has four bands, of which the lowest can be considered as the pinwheel represented in Fig. 4e[30], and $S^z$ can induce internal excitations within this structure, a schematic example being shown in Fig. 4f. To the extent that perturbation theory[47] can be applied with the parameters of SrCu$_2$(BO$_3$)$_2$, the energies of the lowest internal excitations should cover the range from 0 to 1.8 meV, in agreement with the energy range and $|t_+t_+\rangle$ character we obtain (Fig. 4a and c). We comment in passing that our MPS and ED calculations of $S(\mathbf{Q}, \omega)$ cannot be used to distinguish between the $E$ and $A_2$ $|t_+t_+\rangle$ states mentioned above, because this requires a probe of $\Delta S = 2$ rather than $\Delta S = 1$ processes. Unlike the $t_+$ mode, the band centre of the $\Delta S = 1$ internal excitations we observe does not increase linearly with the field beyond $H^*$ (Fig. 4a), which leads to the sublinear form of the low-energy intensity in Fig. 1f. While the position and increasing intensity of this additional scattering contribution suggest an origin for the low-energy intensity observed at 23 and 25.9 T (Figs. 1e and 3a), we stress that the INS intensity is concentrated at energies significantly lower than our MPS results in this regime. The most likely reason for this discrepancy is the fact that our MPS results are obtained on very narrow cylinders, which are not suitable for describing higher densities of condensing pinwheels, and this is why we consider the features of the full spectrum (i.e. including the intermediate and high-energy regimes) to confirm the spin-nematic scenario.

One of the most striking consequences of two-triplon condensation is the presence of low-lying composite three-triplon excitations.

The linear increase of spectral weight away from the $t_0$ and $t_-$ branches is not specific to two-triplon condensation, as several composite two-triplon excitations would be observed if single triplons condense[48]. A leading example is the $|t_+t_0\rangle$ state represented schematically in Fig. 4g, but in a scenario of one-triplon ($t_+$) condensation this would appear in $S^{zz}(\omega, H)$, whereas our discovery of low-energy $|t_+t_0\rangle$ weight in $S^{xx}(\omega, H)$ (Fig. 4b and d) underlines its origin in $S = 2$ bound-state condensation. Quite simply, this scenario makes the number of excitations accessible by INS much larger, and adding three-triplon composites results in the complex spectra we observe both in INS and MPS. Figure 4h shows a schematic representation of a $|t_+t_+t_-\rangle$ composite obtained by exciting a $t_-$ triplon in close proximity to a $|t_+t_+\rangle$ bound state, where in general we find that the low-$S$, low-$S_z$ members of the multiplet benefit from sizeable net binding energies. The bound-state population is reflected in the magnetization, and hence the intensity of the two- and three-triplon excitations is also expected to increase linearly with the applied field, as we observed in Fig. 3d.

## Discussion

To summarize, we have combined high-field INS experiments and cylinder MPS calculations to reach the unambiguous conclusion that, in the field range between $\mu_0H^* = 21$ T and $\mu_0H_c = 27$ T, the ground state of SrCu$_2$(BO$_3$)$_2$ is not a condensate of single triplets, but a condensate of two-triplon bound states. The consequence of this result for the ground state is a broken symmetry corresponding not to magnetic order but to nematic order. The consequences for the excitation spectrum, summarized in Fig. 5a and 5c, are a distinctive one-triplon spectrum with altered gradients beyond a premature kink at $H^*$, a one-triplon gap that persists at all fields and a high field-induced spectral weight of composite two- and three-triplon excitations both internal and external to the two-triplon bound states. We have shown that the weak DM interactions in SrCu$_2$(BO$_3$)$_2$ play no qualitative role in

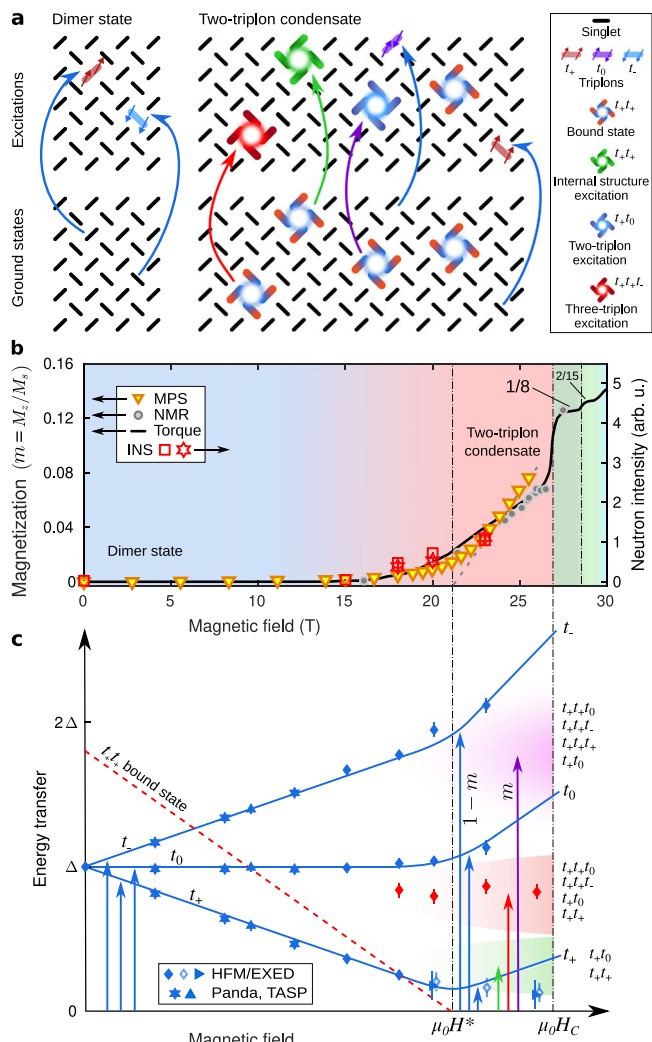

**Fig. 5 | Summary of field-induced static and dynamical magnetic properties.**
**a** Schematic representations of the ground states (lower panels) and associated excitations (upper panels) at low fields in the dimer-product phase (left) and in the spin-nematic phase (right). **b** Magnetization calculated by MPS in the field regime below the 1/8-plateau and compared with results obtained from NMR and magnetic torque measurements (left axis); for comparison we show the measured intensity of the multi-triplon excitations below $t_0$ (right axis). Reprinted with permission from Refs. 12 and 14 (Copyright (2023) by the American Association for the Advancement of Science and the American Physical Society, respectively). **c** Schematic illustration of the field-induced evolution of the primary spectral features identified by combining our neutron spectroscopy data with our MPS and ED results. $\Delta = 3.0$ meV is the one-triplon energy gap at zero field. $\mu_0 H^* = 21$ T is the condensation field for two-triplon bound states (red dashed line). Solid blue lines represent the three one-triplon branches, which show DM-induced rounding of a kink at $H^*$ whose change of gradient reflects $S = 2$ condensation. Green, red and magenta shading represent the field-induced increase of spectral weight in novel multi-triplon excitations of the types represented in Fig. 4(**f**–**h**) and panel (**a**).

determining the physics of the excitation spectrum, but note nevertheless that they do affect the appearance of our results by creating a broad crossover regime below $H^*$. Figure 5b compares the magnetization computed from the MPS ground state with different experimental measurements, all of which indicate that the crossover from near-zero to linearly increasing magnetization spans the range from 16 to 21 T, consistent with Figs. 2h-k and 3d.

As already noted, the bound-state condensate is an example of a spin nematic[39,40], a phase defined by a spontaneously broken rotational symmetry around the magnetic field, and with an order parameter that

is not simply a transverse magnetization but a two-spin operator that does not break time-reversal symmetry. Spin-nematic phases have been predicted to occur just below the saturation field in systems where mixed ferro-antiferromagnetic interactions lead to magnon bound states[49–51], and some evidence for this phenomenon has been obtained both by high-field NMR in LiCuVO$_4$[52] and by high-field calorimetry in Cu$_3$V$_2$O$_7$(OH)$_2$.2H$_2$O (volborthite)[53]. What is new in SrCu$_2$(BO$_3$)$_2$ is that the spin-nematic phase occurs below the first magnetization jump, i.e. at low fields, with dilute triplons binding in a sea of singlet dimers. One primary hallmark of this phase is the gap to one-triplet excitations that persists at all fields and another is the presence of low-lying $S = 2$ excitations around $H^*$. While NMR or INS provide experimental access to the one-triplet excitations, the weak DM terms ensure that ESR can couple to some of the $S = 2$ excitations (at $\mathbf{Q} = (0,0)$). We note that high-field Raman spectroscopy, which is sensitive primarily to $S = 0$ states and $\Delta S = 0$ processes, has also provided a complementary view of the two-triplon condensation process through the behaviour of those excitations[54].

The broken rotational symmetry of the spin-nematic phase should result at finite temperatures in a Kosterlitz-Thouless transition in an effectively 2D system or a 3D XY transition in three. While the full symmetry-breaking would differ depending on whether the $E$ or $A_2$ bound state condenses[38], in SrCu$_2$(BO$_3$)$_2$ the weak DM interactions are likely to change the universality class of this transition to Ising, again complicating a distinction between these two possibilities. Although a transition reported above 26 T[55] has been attributed to this physics, no such observation has yet been made in the field range of 21-26 T that we identify from our INS experiment. This apparent discrepancy might be due to the fact that, as the condensation field of 21 T is approached, the relevant transition temperature decreases below the base value of 500 mK probed to date. This challenging open problem offers a route to deeper understanding of the spin-nematic phase in SrCu$_2$(BO$_3$)$_2$ by future thermodynamic or NMR investigations.

The spin-nematic phase has a close analogy with the physics of superconductivity. In a superconductor, Cooper pairs of fermions undergo a BEC, resulting in a state with a single-particle gap and collective excitations. In SrCu$_2$(BO$_3$)$_2$ and the SSM, the single particle is a triplon, a bosonic particle with an infinite on-site repulsion (a hard-core boson). The spin nematic realized in SrCu$_2$(BO$_3$)$_2$ beyond 21 T can be seen as the BEC of Cooper pairs of bosons. This analogy establishes a direct connection between the superconducting gap of BCS theory and the one-triplon gap in SrCu$_2$(BO$_3$)$_2$, which is equal to half the binding energy at the condensation field ($H^*$). Building further on the analogy, the BCS-BEC crossover[56] links superconductors with large Cooper pairs that extend over many lattice sites to those with compact pairs, both developing coherence through the condensate. In SrCu$_2$(BO$_3$)$_2$, the pairs are the strongly localized pinwheel objects of Fig. 4e and the system represents the extreme BEC limit.

A further analogy is that unconventional superconductivity generically competes with other instabilities of the interacting Fermi sea, such as magnetic and charge order. In SrCu$_2$(BO$_3$)$_2$, the spin nematic is a condensate of pinwheels while the 1/8 magnetization-plateau phase that is favoured above $H_c$ is a crystal of pinwheels, which can be considered as a bosonic CDW order. This transition must be some kind of density-driven Wigner crystallization (or CDW formation) of pinwheels, where the kinetic energy of mobile pinwheels that helps to stabilize the spin-nematic phase is sacrificed to optimize the potential energy of these mutually repelling entities when their packing becomes sufficiently dense. In general this transition could be either continuous or first-order, but the jump in magnetization in SrCu$_2$(BO$_3$)$_2$ (Fig. 5b) indicates clearly that it is strongly first-order. Here we note that the nature of this transition remains an open question in the SSM: the tensor-network approach that identified the crystal of pinwheels in the 1/8 plateau was not able to provide a conclusive result at lower magnetizations[30]. The MPS technique we

employed here used narrow cylinders in order to focus on the spectral function, and hence is not among the methods appropriate for discussing the energetics of pinwheel crystallization.

Finally, our results represent new physics found by combining new capabilities in high-field neutron scattering with state-of-the-art numerical methods, and thus have direct implications in a number of disciplines. They demonstrate the profound importance to correlated condensed matter of scattering facilities able to operate at high magnetic fields, and on the technical side the HZB facility provided valuable information for future generations of high-field experiments. They illustrate directly the value of modern MPS methods in overcoming the long-standing barrier of computing dynamical properties in frustrated systems. For quantum matter, our results reinforce the message that novel composite states and novel forms of many-body order remain to be found at the confluence of strong frustration and controlled extreme conditions.

## Methods

### Neutron spectroscopy

The inelastic neutron scattering (INS) experiment was performed at the High-Field Magnet (HFM) facility of the HZB. The Extreme Environment Diffractometer (EXED) was a time-of-flight neutron instrument that could be operated either in diffraction and small-angle-scattering modes or as a direct-geometry spectrometer. It was designed specifically to function optimally in combination with the HFM, which provided horizontal magnetic fields up to 25.9 T. This magnet was a hybrid solenoid with a 30° conical opening, which, as discussed in Sec. S1 of the SI, placed a considerable restriction on the accessible region of reciprocal space, although rotating the magnet in its entirety with respect to the incoming beam could be used to extend the **Q**-coverage.

The sample was a single-crystalline rod with a mass of 2.5 g, which was grown by a floating-zone method with its (100) direction orientated approximately along the growth axis[57,58]. The magnetic field was applied along the (001) direction in the horizontal scattering plane, a geometry chosen to give access to part of the $(Q_h, Q_k, 0)$ plane and to avoid the combination of the applied magnetic field with the DM interactions breaking any further symmetries of the system. The incident neutron energies were $E_i = 4$ and 8 meV and the rotation angle of the magnet relative to the incoming neutron beam was $\varphi = -10°$. A dilution refrigerator purpose-built by HZB in collaboration with the University of Birmingham allowed the measurements to be performed at 200 mK. Data were collected for periods between 7 and 20 hours for each selected field strength with $E_i = 8$ meV and between 10 minutes and 6 hours with $E_i = 4$ meV.

Supporting INS data were collected on the triple-axis spectrometers TASP, at the Paul Scherrer Institute, and Panda, at the Heinz Maier-Leibnitz Zentrum. On TASP, the same single crystal was orientated with $(Q_h, Q_k, 0)$ in the horizontal scattering plane and a vertical magnetic field of 9.5 T was applied. The measurement was performed at 2.3 K, with a constant final neutron momentum $k_f = 1.5$ Å$^{-1}$ and with a liquid-nitrogen-cooled Be filter to suppress the $\lambda/2$ contribution. On Panda, measurements were performed in the same geometry, at 1.5 K and with final neutron momentum $k_f = 1.55$ Å$^{-1}$, for fields of 4, 8 and 12 T. The data collected at 8 T is reprinted with permission from Ref. 22 (Copyright (2023) by the American Physical Society).

### MPS Calculations

Matrix-Product States (MPS) are a variational Ansatz that can be used to represent the wavefunction of 1D quantum systems[59]. The representation is formulated in terms of rank-3 tensors and its accuracy controlled by the tensor bond dimension, $\chi$. A 2D system can be described by wrapping the lattice onto a cylinder and reformulating the model as a 1D Hamiltonian with further-neighbour interactions[60]. The ground state in the MPS representation is then obtained using the Density-Matrix Renormalization-Group (DMRG) algorithm[59,61].

The spin physics in $SrCu_2(BO_3)_2$ is well described by the SSM Hamiltonian with Heisenberg interactions $J = 81.5$ K and $J' = 0.63J$[62], supplemented by DM interactions $D = 0.034J$ on the intra-dimer bonds[34] and $D' = -0.02J$ on the inter-dimer bonds[32]. Finite DM interactions were also required to obtain uniform magnetization distributions in our calculations of the ground state in an applied field. All of the spectral functions shown in the main text were computed with a cylinder of length $L = 20$ and circumference $W = 4$ sites, as shown in Sec. S2A of the SI, where we benchmark the effect of $W$ by computing the energy per site, magnetization distribution and spectral function at two selected fields for a cylinder with $W = 6$.

To calculate the dynamical properties of the system, we employed the time-dependent variational principle (TDVP)[63–65]. This method makes use of Lie-Suzuki Trotterization not of the Hamiltonian but of projectors to the tangent space of the MPS, which are constructed from the MPS with an accuracy dependent on $\chi$. We used the two-site variant of the TDVP algorithm, which allows the bond dimension of the MPS Ansatz to be readjusted as the entanglement grows upon time-evolution, with a time step of $0.16/J$ and keeping a maximum $\chi$ of 600. The excitations of the SSM are localized as a consequence of the strong frustration and thus the time-dependence appears as a slowly expanding time cone. This property allowed us to continue the real-time evolution to rather long times even on short cylinders.

In the magnetized system, $\langle S^z \rangle$ is uniform while $\langle S^x \rangle$ and $\langle S^y \rangle$ are staggered, and hence the magnetic unit cell contains $n_s = 4$ sites. To compute the dynamical structure factor, we therefore started from each of these sites individually and took the Fourier transform

$$S^{\alpha\tilde{\alpha}}(\mathbf{k},\omega) = \frac{1}{n_s} \sum_{a,b} e^{-i\mathbf{k}\cdot(\mathbf{x_a}-\mathbf{x_b})} S_{ab}^{\alpha\tilde{\alpha}}(\mathbf{k},\omega), \quad (1)$$

where $\mathbf{x}_a$ and $\mathbf{x}_b$ are the relative positions of sites $a$ and $b$ within the unit cell and

$$S_{ab}^{\alpha\tilde{\alpha}}(\mathbf{k},\omega) = \frac{1}{N} \sum_{\mathbf{R}} \int_{-\infty}^{\infty} e^{i(\omega t - \mathbf{k}\cdot\mathbf{R})} \langle S_{a,\mathbf{R}}^\alpha(t) S_{b,\mathbf{0}}^{\tilde{\alpha}}(0) \rangle dt, \quad (2)$$

with $N$ the number of unit cells, $\mathbf{R}$ the spatial position of each cell and spin components $\alpha = \tilde{\alpha} \in \{z, x\}$. To obtain the results shown in Figs. 1f and 4a, b, we integrated these functions over the full reciprocal space.

## Data availability

The data that support the findings of this study are available at https://doi.org/10.5281/zenodo.8434526 and from E.F. upon request. Source data are provided with this paper.

## Code availability

The code producing the results in this study is available from M.N. upon request.

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

## Acknowledgements

We are especially grateful to S. Gerischer, P. Heller and R. Wahle for their assistance with cryogenics and to S. Kempfer and P. Smeibidl for ensuring the operation of the high-field magnet. We thank J. Pásztorová and L. Testa for their assistance with sample preparation. We thank C. Batista, O. Gauthé, L. Herviou, B. Kumar, A. Läuchli and S. Miyahara for helpful discussions. We acknowledge the financial support of the European Research Council through the Synergy network HERO (Grant No. 810451) and of the Swiss National Science Foundation through Project Grant No. 188648. We are grateful to the Helmholtz-Zentrum Berlin for the allocation of neutron beam time on EXED, to the Paul Scherrer Institute for beam time on TASP and to the Heinz Meier-Leibnitz Zentrum for beam time on Panda. Our numerical calculations were performed on the facilities of the Scientific IT and Application Support Center of the EPFL. This publication was made possible in part by the generous support of the Qatar Foundation through the Seed Research Funding Program of Carnegie Mellon University in Qatar. The statements made herein are solely the responsibility of the authors.

## Author contributions

The experimental project was conceived by E.F., H.M.R. and K.K., with input from H.N. and H.K. The theoretical framework was provided by F.M. Single crystals were grown by E.P. INS measurements on HFM/EXED were performed by E.F., K.M., K.K., O.P. and M.B., on TASP by E.F., J.-R.S. and A.A.T. and on Panda by M.E.Z. Data analysis was performed by E.F., M.E.Z. and H.M.R. MPS and ED calculations were performed by M.N. and their theoretical interpretation was provided by M.N., B.N. and F.M. The manuscript was written by E.F., H.M.R., M.N., B.N. and F.M. with contributions from all the authors.

## Competing interests

The authors declare no competing interests.
