## [Peer Review File · Nature Communications]

Field-induced bound-state condensation and spin-nematic phase in $\text{SrCu}_2(\text{BO}_3)_2$ revealed by neutron scattering up to 25.9 TREVIEWER COMMENTS

Reviewer #1 (Remarks to the Author):

The manuscript reports the measurement of low energy magnetic excitations in the geometrically frustrated quasi-two dimensional quantum magnet, $\text{SrCu}_2(\text{BO}_3)_2$, at magnetic fields up to 25.9 T using inelastic neutron scattering studies at a purpose-built, high field facility. The experiments are complemented by state-of-the-art Density Matrix Renormalization Group (DMRG) simulation of the underlying microscopic Hamiltonian using a Matrix Product State (MPS) representation. By comparing the results from numerical simulations with experimental data, the authors claim that the ground state of $\text{SrCu}_2(\text{BO}_3)_2$ in the field range 21-27 T is a unique condensation of two-triplon bound states in their $|S=2, S_z=2\rangle$ spin configuration. The bound states resonate between two orthogonal arrangements locally leading to a nematic, rather than magnetic ordering.

The experimental data are of high quality. Performing accurate measurements in INS experiments at high magnetic field is challenging, but the authors have been able to obtain clean data with sufficient energy resolution. The experiments will add greatly to our understanding of this unique quantum magnet. The numerics are performed using well-established techniques and there is no reason to doubt the validity of the calculations. The authors have been thorough in their exploration of the different parameter regimes and have calculated all the experimentally relevant observables to characterize the ground state phases. However, in the opinion of the present referee, the interpretation of the experimental data in terms of the numerical simulation has some ambiguity that needs to be resolved before the manuscript can be considered for publication:

1. The central result of the paper is captured in Fig.1, panels (e) and (f) where the experimental data is compared with results from numerical simulations. While the numerical simulation reproduces most of the experimental observations, there are some crucial differences that relevant to the claims made. The experimental data do not exhibit any increase in intensity at low energy transfers at fields above 20 T. This is in contrast to the simulation results that show a steady increase in intensity in the 21-25 T range. The interpretation of the ground state as a condensation of $S=2$ bound states depend crucially on this increase. The authors have claimed that the noise floor of the instrument prevents them from exploring this regime, but a quick comparison with the simulation data shows that the noise floor is sufficiently low that signatures of any such feature should be observable. It is also observed that there is a mismatch in the high energy spectrum at fields above 15 T – a significant fraction of the spectral weight is transferred to higher energy transfers. This underscores the fact that the microscopic model does not completely capture the field induced excitations. While disagreement in high energy properties between experimental observations and predictions from microscopic model designed to describe ground state properties is usually not fatal (after all, no microscopic model can quantitatively capture every experimental observation), in this case, this might point towards some significant shortcoming, especially when it explores the field evolution of the low energy excitation modes.

2. The authors mention that a ferromagnetic inter-dimer Heisenberg exchange interaction is needed to obtain the ground state condensation of $S=2$ triplon bound states at magnetic fields above 21 T. But such an interaction is contrary to the widely accepted and well established microscopic model for $\text{SrCu}_2(\text{BO}_3)_2$ that has been used in all previous studies of the material, including some landmark studies by some of the authors. Are the authors challenging this model and proposing a new one in this work? To make matters more confusing, in the methods section and in the Supplementary material, they mention using the standard model for DMRG. It will be good to clarify this issue.

3. Is it possible to estimate the moment of the elementary quasiparticles (isolated triplons or bound states) from the slope of the magnetization curve?

4. How do the low-field results of the present study compare with previous INS studies of $\text{SrCu}_2(\text{BO}_3)_2$? Are the interpretations in those studies consistent with the current modelling? Such a discussion is sorely missing.

In the opinion of the present referee, the authors need to satisfactorily address the above questions before the manuscript can be considered for publication.

Reviewer #2 (Remarks to the Author):

The manuscript reports a combined theoretical and experimental study on the spin excitations of $\text{SrCu}_2(\text{BO}_3)_2$. The authors conducted experimental measurements of the excitation spectrum using inelastic neutron scattering at 200 mK with magnetic field strengths up to 25.9 T. Furthermore, they performed matrix-product-state calculations to determine the dynamical structure factor of the Shastry-Sutherland model, complementing the experimental findings. By thoroughly analyzing the experimental and theoretical results, they provide compelling evidence that in the field range between 21 T and 25.9 T, the ground state of $\text{SrCu}_2(\text{BO}_3)_2$ is a condensate of two-triplon bound states, which forms a spin nematic state.

As a typical frustrated quantum magnetic system, $\text{SrCu}_2(\text{BO}_3)_2$ and its closely related Shastry-Sutherland model are ideal platforms for studying quantum phase transitions. In particular, the topic of magnetic-field-induced quantum phase transitions and magnetization-plateau phenomena is highly interesting for the quantum magnet community. This research yields novel findings in the crossover region preceding the $1/8$ magnetization-plateau phase, accompanied by a thorough analysis of its underlying physical mechanisms. I think the question addressed in this manuscript is important, and the results being reported represent a significant advance. However, before I can recommend this work for publication, the authors should address the following concerns:

1) In the "Methods" section, the authors elucidate that they have used a single-crystalline sample in the experiments. In principle, this can enable them to acquire the evolution of excitation spectra in momentum space, consequently demonstrating the dispersion characteristics of different excitation modes. However, in Fig. 2a-f, they showed the excitation spectra by integrating in two directions instead of presenting the original distribution of spectral functions in momentum space. This compresses the excitation spectrum into one dimension, making it challenging to discern the dispersion information associated with different excitation modes. Why don't they show the original distribution of the spectrum in momentum space?

While the authors claim that the dispersion of spin excitation is weak, exhibiting the original excitation spectrum would offer a more lucid illustration of this assertion. Moreover, in Fig. 2a-f, the distribution of excitation spectra in the Q_h direction appears non-uniform. Thus, comparing the momentum dependence of the spin excitation spectra would better demonstrate the consistency between experimental and theoretical results. I would prefer to see the authors to compare the excitation spectra at some typical momentum points in both experimental and theoretical data, rather than relying solely on the integrated results. This may better illustrate whether the theoretical explanation is truly applicable to this system.

2) What is the spectrum with negative energy in Fig. 3(a), and How it was obtained? Additionally, where are the solid blue and red lines mentioned in the caption of Fig. 3(a)?

3) How does the nematic phase, resulting from a condensate of $s=2$ two-triplon bound states, evolve into the $1/8$ magnetization-plateau phase, which is a crystal of $s=2$ two-triplon bound states? It would greatly benefit the readers if the authors can give more detailed explanation about the underlying mechanisms involved in this evolution.

Reviewer #3 (Remarks to the Author):

The authors reported inelastic neutron scattering (INS) studies of the SSL $\text{SrCu}_2(\text{BO}_3)_2$ under magnetic fields up to 25.9 T, near the first $1/8$ plateau phase of the SSL. Through MPS calculations, they discovered an exotic spin-nematic phase that arises from the condensation of $S=2$ bound states, which provides the best explanation for the unconventional features observed in their INS experiments. This finding is novel and represents a significant advancement in our understanding of triplon excitations in the SSM. Both the experimental and theoretical modeling approaches are robust and represent the forefront of research in these fields. I recommend publication of this work after addressing the following points.

- 1) Figure 1 includes a lot of details. It is important to make the symbol shape easier to be tell. There are many colors and types of lines and please check the description of each in the caption. "e)" may be missing in the caption.
- 2) In plots Fig. 1e and 1f, both the INS data and MPS calculations show gradual upturns above 21 T, while the black dashed lines exhibit much sharper turns. It is unclear what the dashed lines represent and why they deviate from the INS and MPS data.
- 3) The dispersion within each excitation band has been overlooked. How does the variation in dispersion under the magnetic field affect the unconventional observations in the INS, such as the gap position and the definition of gap closure? High-resolution cuts in different Brillouin zones (BZs) from the theoretical calculations could provide insights into these questions and offer stronger support for the claims in this work.
- 4) Regarding the bosonic pairs, can they be spatially separated or have to be bonded like the two-triplon bound state shown in Fig4e? It seems that there is no energy cost to spatially separate two triplets. The state/excitation needs to be paired but two triplets does not need to be bonded together. Is this correct?

Letter to Reviewers

We are grateful to the reviewers for their comments which have allowed us to further improve our manuscript. Please find below a list addressing the questions that were raised.

Reply to the Report of Reviewer #1

We would like to thank the reviewer for a critical reading and an accurate summary of our work, as well as for their praise for the quality and rigour of our experimental and numerical analysis. The reviewer's final opinion, that "the interpretation ... has some ambiguity that needs to be resolved before the manuscript can be considered for publication," is based primarily on the first of the four points to follow.

1.1) The reviewer states that there are some "crucial differences" between our INS and MPS results in Figs. 1e and 1f, and questions whether these could invalidate our entire interpretation. Dealing first with the easier case of high-energy mismatches, we agree in full with the reviewer's comment that "the microscopic model does not completely capture the field induced excitations," and stated this ourselves in a number of places in our manuscript. We stress again that most of our MPS calculations are performed on a cylinder of width $W = 4$ (benchmarked in the SI against $W = 6$) and thus we make no claims whatsoever to full quantitative accuracy. We thank the reviewer for the observation "after all, no microscopic model can quantitatively capture every experimental observation," and add that, while we have used the standard J , J' , D and D' parameters in a 2D SCBO model (see also below), the real material also has 3D coupling, incompletely specified D' interactions and magnetostriction. We suggest that a quantitative error of 0.5 or 1 meV in any given feature when $J = 7$ meV cannot be ranked as a "significant shortcoming."

Turning to low energies, here a quantitative error of 0.5 meV certainly does appear as a dramatic discrepancy. Again we fully agree with the reviewer that the low-energy sectors of Figs. 1e and 1f above 20 T cannot be used as stand-alone verification of the spin-nematic scenario, and we apologise if our presentation gave this impression: certainly we intended that our logic requires (i) the higher-energy features of the spectrum and (ii) the detailed separation of spectral contributions presented in Figs. 2 and 4 in order to reach our conclusions. We have reviewed our presentation in this respect and have tried to make this logic more transparent. Turning to items of physics, the reviewer is correct that our INS data at 20, 23 and 26 T do reach energies low enough to show that the "internal structure" contribution of the $|2,2\rangle$ band of SCBO lies lower than in $W = 4$ MPS. On the data side, we have revised our fits and we have been slightly less conservative in our attribution of signal, as a result of which we have included some data at lower energies in the revised Fig. 1e. Still the MPS data lie higher than the INS data, and we use them primarily for the qualitative purpose of understanding the overlapping one- and two-magnon contributions: while we expect the one-magnon part to rise with a slope of 1 (depicted by the dashed line), the two-magnon part is relatively flat and gains intensity as the field is increased. We were uncertain from the reviewer's wording as to whether they found a discrepancy in energy or intensity in this sector, but we find the qualitative features, including when separated into the S_{zz} and S_{xx} contributions, to be fully plausible and have discussed this more clearly around our analysis of Fig. 4a.

As we discussed in some depth in the manuscript, we also did not accept the level of agreement in the low-energy sector alone as a proof of the spin-nematic scenario, and the strength of our contribution is to provide a consistent picture based on the majority of the magnetic spectrum. Thus we analysed the intermediate- and high-energy one-triplon branches, and their surrounding energy windows, to gain the insight required to

understand the physics. Crucial here is that, if the kinks in the one-triplon branches occur at 24 instead of 21 T, as required by a $|1,1\rangle$ condensation scenario, one finds a dramatically poorer fit to the t_0 and t regions, even when quantitative errors of 0.5 to 1 meV are considered. Figure 2k provides a particularly clear example where the t_0 branch has already moved significantly upwards at only 23 T. Further, our statements are based not only on the one-triplon branches but also on the weights and weight-shifts we observed (both by INS and by resolving our MPS results into their transverse and the longitudinal parts) between the one- and higher-triplon contributions. We hope that the reviewer can accept the totality of the picture in Figs. 1-4 in lieu of a point-by-point quantitative agreement between the lower right corners of Figs. 1e and 1f. In the revised manuscript we have extended the overall discussion in an effort to walk the reader through the experimental and numerical evidence more carefully.

1.2) The reviewer states “The authors mention that a ferromagnetic inter-dimer Heisenberg exchange interaction is needed to obtain the ground state condensation of $S=2$ triplon bound states at magnetic fields above 21 T.” This is completely false. We apologise for the aspects of our presentation that led to this impression. We are using precisely “the widely accepted and well established microscopic model for $\text{SrCu}_2(\text{BO}_3)_2$ that has been used in all previous studies of the material” and we are absolutely not “challenging this model and proposing a new one” in the present work. The statements we made in the main text and SI about using the standard model are completely accurate and we apologise for assuming that they were sufficient. The origin of this misunderstanding lies in the toy model we used in Figs. 1c and 1d: we stress that this is an absolutely minimal model that produces the condensation of a $|2,2\rangle$ state and we did not wish to claim any connection between this and SCBO (the toy geometry is entirely wrong). We introduced the toy model for the purely conceptual purpose of illustrating the fact that, if the $|2,2\rangle$ state condenses before the $|1,1\rangle$ state, then the $S = 1$ branches have a very characteristic change of gradient (as a function of the field) by 2 units instead of 1, and the $|1,1\rangle$ gap is always finite.

We have taken a number of steps to remove the misleading impression that our former presentation gave to the reviewer: we have emphasised further in the main text that (i) we use the standard model for SCBO and (ii) the toy model in Figs. 1c-d is truly a conceptual toy, and barely a model. We have rearranged Fig. 1 to move the toy illustration further from the SCBO illustration and we have relabelled the interaction parameters of the toy model to avoid confusion with parameters of the SCBO model. We hope that these steps are sufficient to avoid any misunderstanding on the part of the reader and we apologise again to the reviewer for this situation.

1.3) The reviewer asks if the magnetic state of the condensing quasiparticles is reflected in the gradient of $M(H)$. Unfortunately this is not the case. Only in a very small-system calculation could one distinguish between the two possibilities (one-triplon condensation versus spin-2 bound-state condensation) by keeping track of the size of the finite steps in the magnetization: $\Delta S = 1$ versus $\Delta S = 2$. In a macroscopic experiment, the gradient is determined by the dependence of the energy on the magnetic field, and this can a priori take any value regardless of whether the microscopic magnetic particles have spin 1 or 2.

1.4) Our low-field results are those of previous INS studies (we did provide the references for the data we show at 4, 8, 9.5 and 12 T). They are excellently described by the standard model of SCBO and for this reason there was no discussion to present – we have no new results here. This question of the reviewer was caused by the misunderstanding in point (1.2) and we apologise again for that.

We thank the reviewer once again for helping us to clarify our presentation at several key places in the manuscript.

Reply to the Report of Reviewer #2

We are grateful to the reviewer for a detailed appraisal of our work and for the opinion “I think the question addressed in this manuscript is important, and the results being reported represent a significant advance.” The reviewer raises three points of concern:

2.1) The reviewer is correct that it is always preferable to have extensive spectral information over the entire Brillouin zone (BZ). In our present experiment, the presence of the high-field magnet caused a very dramatic restriction of the accessible region of reciprocal space, as we discussed in detail in Sec. S1 of the SI. This made it impossible to achieve the same type of BZ coverage as that at the disposal of authors working previously at lower fields, and also restricted us to a region where the structure factor is at its lowest (Fig. S2).

However, the status of research on SCBO and the properties of the material are such that this restriction was not a fundamental issue. First, although the reviewer writes “This may better illustrate whether the theoretical explanation is truly applicable to this system.”, the type of information this process would reveal concerns the parameters of the spin Hamiltonian. Here we do not call into question the literature values of the J , J' , D and D' parameters (referred to by Reviewer #1 as the “standard model of SCBO”), and use precisely the terms fitted accurately in previous work. The nature of this (adapted Shastry-Sutherland) model is that the one-triplon excitations are almost completely flat throughout the BZ, with only a very small (0.3 meV) dispersion arising from the DM terms. This is the primary reason that we integrate over all of Q : the nature of SCBO is such that there is very little information to lose. Certainly within the region of Q -space covered by HFM/EXED, all Q points show an almost identical form of the intensity as a function of energy transfer, and even the ability to reach the BZ corner studied in earlier work would not change this. To illustrate this situation, we refer to the figure below, which was computed by MPS for cylinders with the standard (J , J' , D , D') parameters using cylinders width $W = 4$: it is clear that the modes are entirely flat in Q_h at all energies and for all fields, fully justifying the integration. Only the intensity pattern as a function of Q at fixed energy might show a richer structure (Fig. S10), were we able to access higher Q .

The reviewer is certainly correct that the spectra shown in Figs. 2a-f are non-uniform, but this effect lies almost entirely in the intensity, as Fig. S6 (lower two rows) and the figure presented here make clear. SCBO is a fully 2D material, and the equivalence of the a' and b' crystallographic axes already justifies much of the data integration we performed. We repeat that a one-triplon dispersion is neither expected in SCBO nor is it relevant to the

formation of the spin nematic, because the gap of this branch never closes. [While the dispersion of the $|2,2\rangle$ branch would be very valuable information, neither can INS probe $S = 2$ processes and nor could we use HFM/EXED to access the energy range relevant for the “internal structure” excitations, which are manifest only at fields above 20 T.] We trust that the reviewer can accept that comparing different Q values does not bring the extra information that it does in most (imperfectly frustrated) magnetic materials, and we have added some words in the revised manuscript to emphasise this point. Certainly if the reviewer does feel that access to the higher-intensity, higher- Q points out to the edge of the BZ is important for the understanding of SCBO then HFM/EXED was not the tool for the task and our work will not be able to satisfy this criterion.

2.2) The spectrum with negative energy transfer represents processes where the neutron receives energy from the system and was measured on HFM/EXED in the same way as the positive energy transfers. However, at the extremely low temperatures we consider, where no quasiparticles are excited, it is understandably flat and can be used to obtain more information about the background (scattering processes in the signal but not due to the sample).

Concerning the “solid red and blue lines,” this was an error in the figure preparation where we forgot to implement the final update. We apologise for this, and have now marked the observed data with solid lines as stated in the caption, while the extracted excitation is shown by the dashed lines.

2.3) The reviewer poses an excellent question concerning the evolution of the spin nematic into the $1/8$ -plateau phase. Indeed both involve $|2,2\rangle$ two-triplon bound states (“pinwheels”) and thus this transition must be some kind of density-driven Wigner crystallisation (or CDW formation) of pinwheels, where the kinetic energy of mobile pinwheels that helps to stabilise the spin-nematic phase is sacrificed to optimise the potential energy of these mutually repelling entities when their packing becomes sufficiently dense. On general grounds, such a transition could a priori be continuous or first-order. It turns out that, in SCBO, this transition is strongly first order, as demonstrated by the jump in magnetization before entering the plateau. On the theory side, while the nature of the $1/8$ -plateau phase could be unambiguously identified with tensor networks, the nature of the transition into that phase could not be clearly determined (see Ref. 32). Our present work, with its focus on the excitation spectrum and its consequent restriction to very small cylinders, is not suited to answering this type of question, and thus we have no new insight to offer. In the revised version of the manuscript, we have added a short paragraph that summarises this qualitative point and have repeated the most relevant reference. We hope that these changes provide the appropriate service to the reader concerning this point that the reviewer had in mind.

We would like to thank the reviewer once again for these comments, which have allowed us to improve our presentation in several places.

Reply to the Report of Reviewer #3

We thank the reviewer for a concise and insightful critique of our manuscript and for a recommendation in favour of publication, subject to clarifying the following four points:

3.1) We are grateful to the reviewer for this comment. We assume it refers primarily to Fig. 1e (the e label is in fact present in the caption). We prepared many versions of this figure and perhaps lost sight of its readability. We have made a number of changes to this figure panel, including as a result of the comments of Reviewer #1: we have extended some of the coloured regions, we have introduced a system of symbol sizes to improve the prioritisation and we have paid particular attention to the labels and descriptions in the caption. We hope that these changes lead to the desired improvements.

3.2) In Figs. 1e and 1f, the dashed lines are literally a “guide to the eye.” They are not based on any type of fit, only on the schematic concept illustrated in Figs. 1c and 1d that, if a model allows the field-induced condensation of an $S = 2$ mode, then all the excitation branches will show a discontinuous gradient with a step of 2 instead of 1 [see response (1.2)]. The dashed lines show this result not for a toy model of 2 dimers (Figs. 1c-d) but for a Shastry-Sutherland model with the J and J' parameters of SCBO but D and D' set to zero. While the kink we wish to highlight is indeed rounded out in SCBO (i.e. in our INS and MPS results) because of the DM terms, the dashed lines do a surprisingly good job of tracing the strong one-triplon intensity. The single most important result of these considerations is our ability, based on all 3 branches, to trace the kink to approximately $H^* = 21$ T, a field that ensures consistency between expectation and the majority of our observations. Here we stress in addition that the strongly increasing two-triplon intensity above 20 T is responsible for the fact that the low-energy dispersion at these fields is quite flat, rather than reflecting the linear increase of the one-triplon part alone. We have modified the text of the revised manuscript in order to give a more detailed statement of the origin of the dashed lines in question, and later of the physical messages they deliver.

3.3) The reviewer asks about the dispersion of the excitation bands. Here we refer to our reply to the first point of Reviewer #2. The special nature of SCBO is such that there is no dispersion of the one-triplon excitations, as shown in the figure in reply (2.1). The special nature of the HFM/EXED instrument was such that we could not access any higher Brillouin zones of SCBO. We trust that the reviewer can accept that this kind of information is simply unattainable at the target fields of our present experiment, and that in any case it does not provide any additional insight or support for a particular scenario due to the perfectly frustrated nature of SCBO.

The reviewer mentions the possibility of a change in dispersion due to the magnetic field. Because the spin Hamiltonian is predominantly Heisenberg, and only the DM terms do not commute with the field, the modifications caused by the field are extremely small (fractions of 0.1 meV, which certainly could not be resolved on HFM/EXED). The reviewer also asks about changes to the gap position and the definition of gap closure: here it is important to distinguish between the one-triplon branches, which are extremely flat and whose gap remains in the same place in Q without ever closing, and the two-triplon branches. The latter are not so flat and the gap to one of them does close, at one point in Q ; while this point does not change with field for a given band of the $|2,2\rangle$ manifold, the unknown at the time of writing is which band lies lowest, and hence whether the gap closes at $(0,0)$ or (π,π) . As explained in the manuscript; because we cannot probe these bands by INS, we have tried instead to seek the less direct fingerprints of $|2,2\rangle$ condensation in the higher-lying parts of the spectrum.

3.4) The reviewer asks about the spatial nature of the two-triplon bound state. Unfortunately it is not correct that there is no energy cost to separate the two triplons. Their bound state is possible only when they form the pinwheel state depicted in Fig. 4e: without the resonant energy of alternating between the two configurations shown, the $|2,2\rangle$ state is simply not bound (and similarly the $|1,1\rangle$ and $|0,0\rangle$ two-triplon states reach their maximally bound energies in this configuration). This is not our result, and we have referenced the specialist studies of the ground state that revealed it. We trust that our presentation, and particularly Figs. 4 and 5, make sufficiently clear the importance of the pinwheel configuration to optimising the binding energy within the spin-nematic state.

We would like to express our thanks to the reviewer for assisting us, with all of these comments, in enhancing the quality and readability of our manuscript.

REVIEWERS' COMMENTS

Reviewer #1 (Remarks to the Author):

I am satisfied with the responses of the authors to my questions / comments. I believe the manuscript is now suitable for publication.

Reviewer #2 (Remarks to the Author):

The authors have replied to my comments in a very positive way and by making great efforts. I appreciate their effort and understand the experimental restriction in the high-field magnets. Adding a qualitative point relating to the evolution of the spin nematic into the 1/8-plateau will be helpful to readers. Therefore, I recommend the revised manuscript to be published in Nature Communication.

Reviewer #3 (Remarks to the Author):

The authors have addressed all of my questions and concerns. I recommend it to be published as is.

Letter to Reviewers

We would like to thank the reviewers again for all their comments which have allowed us to improve the readability and clarity of our manuscript.

Reply to the Report of Reviewer #1

We thank the reviewer for recommending our manuscript for publication.

Reply to the Report of Reviewer #2

We are grateful to the reviewer for once again reading through the manuscript with a critical eye. We have included the following sentence about the nature of the transition between the spin-nematic phase and the $1/8$ plateau:

“This transition must be some kind of density-driven Wigner crystallisation (or CDW formation) of pinwheels, where the kinetic energy of mobile pinwheels that helps to stabilise the spin-nematic phase is sacrificed to optimise the potential energy of these mutually repelling entities when their packing becomes sufficiently dense.”

This sentence is introduced at the bottom of the 1st column on page 9, in the paragraph discussing exactly this transition and added in the previous round of changes to address the reviewer’s question regarding this topic. Unfortunately, from our MPS calculations, we can not say anything specific about how the spin-nematic phase evolves onto the $1/8$ plateau. The cylinder size is simply too small to accommodate a sufficient number of bound-state objects to obtain the $1/8$ state. The magnetization curve of $\text{SrCu}_2(\text{BO}_3)_2$ strongly suggests that the transition is of 1st order but in the Shastry-Sutherland model this is still an open question.

We would like to thank the reviewer for recommending the publication of the revised manuscript.

Reply to the Report of Reviewer #3

We thank the reviewer for recommending our manuscript for publication.